# An Update of Bovine Hemoplasmas Based on Phylogenetic and Genomics Analysis

**DOI:** 10.3390/microorganisms10101916

**Published:** 2022-09-27

**Authors:** Diana Laura Flores-García, Hugo Aguilar-Díaz, Itzel Amaro-Estrada, Fernando Martínez-Ocampo, Rosa Estela Quiroz-Castañeda

**Affiliations:** 1Departamento de Ingeniería en Biotecnología, Universidad Politécnica del Estado de Morelos, Paseo Cuauhnahuac 566, Lomas del Texcal, C.P., Jiutepec 62574, Morelos, Mexico; 2Centro Nacional de Investigación Disciplinaria en Salud Animal e Inocuidad, Instituto Nacional de Investigaciones Forestales, Agrícolas y Pecuarias, Carretera Federal Cuernavaca-Cuautla No. 8534, Col. Progreso, Jiutepec 62550, Morelos, Mexico; 3Laboratorio de Estudios Ecogenómicos, Centro de Investigación en Biotecnología, Universidad Autónoma del Estado de Morelos, Av. Universidad No. 1001, Col. Chamilpa, Cuernavaca 62209, Morelos, Mexico

**Keywords:** hemoplasmas, molecular detection, antigens, genomics

## Abstract

*Mycoplasma wenyonii* and ‘*Candidatus* Mycoplasma haemobos’ are bacteria that have been described as significant hemoplasmas that infect cattle worldwide. Currently, three bovine hemoplasma genomes are known. This work aimed to describe the main genomic characteristics and the evolutionary relationships between hemoplasmas, and provide a list of epitopes predicted by immunoinformatics as diagnostic candidates for bovine hemoplasmosis. Thus far, there is no vaccine to prevent this disease that economically impacts cattle production worldwide. Additionally, there is a lack of vaccines against bovine hemoplasmosis. In this work, we performed a genomic characterization of hemoplasmas, including two Mexican strains reported in bovines in the last few years. The generated information is a new scenario about the phylogeny of hemoplasmas. Also, we show genomic features among hemoplasmas that strengthen their characteristic genome plasticity of intracellular lifestyles. Finally, the elucidation of antigenic proteins in Mexican strains represents an opportunity to develop molecular detection methods and diagnoses.

## 1. Introduction

Hemotrophic mycoplasmas (hemoplasmas) are a group of erythrocytic bacteria pathogens of the *Mollicutes* class that infect a wide range of vertebrate animals [1,2]. At first, these small and uncultivable in vitro bacteria were classified as genera *Haemobartonella* and *Eperythrozoon* within the *Anaplasmataceae* family and *Rickettsiales* order [2]. However, the genetic analysis of the 16S ribosomal RNA (rRNA) gene and morphologic similarities showed that these bacteria are closely related to the *Mycoplasma* genus [1,3]. In 2001, a formal proposal was presented to transfer these organisms to the genus *Mycoplasma* within the *Mycoplasmataceae* family [4]. Currently, 12 hemoplasma genomes have been identified in the GenBank database (https://www.ncbi.nlm.nih.gov/genbank/, accessed on 15 August 2022), including *Mycoplasma (M.) wenyonii* strains: Massachusetts and INIFAP02, and one *Candidatus* (*Ca.)* Mycoplasma haemobos strain, INIFAP01 [5,6,7]. These hemoplasma genomes provided relevant information about possible pathogenic mechanisms, metabolism, and divergence compared to other mycoplasma species [1].

Currently, *M. wenyonii* and *Ca*. M. haemobos are significant hemoplasmas infecting cattle worldwide [8,9,10]. In cattle, acute hemoplasma infections are rare and characterized by anemia, fever, depression, and diarrhea [11,12]. Chronic bovine hemoplasma infections are associated with variable clinical signs (that is often confused with clinical signs of diseases such as anaplasmosis), including low-grade bacteremia, weight loss, decreased milk production, reduced calf birth weight, pyrexia, scrotal and hind limb edema, infertility, and reproductive inefficiency. Consequently, bovine hemoplasmas cause significant economic losses worldwide, mainly when associated with pathogens of the genus *Anaplasma* (bacteria) or *Babesia* (protozoan) [8,13,14]. In addition, latent and asymptomatic infections have also been reported for hemoplasmas [14], and single infections or coinfections of *M. wenyonii* and ‘*Ca*. M. haemobos’ [15,16,17]. 

Genome sequencing is an alternative to overcome the difficulties of culturing and studying hemoplasmas. Currently, reports of genomic characterization of bovine hemoplasmas are scarce. However, three genomes are reported worldwide: *M. wenyonii* strain Massachusetts from the United States [5], *M. wenyonii* INIFAP02 [7], and ‘*Ca*. M. haemobos’ INIFAP01, from Mexico [6]. 

On the other hand, immunoinformatics is an effective and powerful tool that helps predict and identify antigenic and immunogenic epitopes in proteins for several purposes and applications. Also, in silico analysis allows the processing of a significant amount of data in a short time [18,19,20].

In this work, we carry out a pangenome characterization of hemoplasmas, including the Mexican strains reported in cattle. Based on phylogeny, we present a new scenario about how hemoplasmas organize into two groups. Furthermore, we show genomic features that strengthen the plasticity of hemoplasmas as intracellular bacteria. Finally, elucidating some antigenic proteins in Mexican strains represents an opportunity to develop diagnosis and molecular detection methods.

## 2. Materials and Methods

### 2.1. Genome Sequences and Annotation

In Mexico, the Anaplasmosis Unit (CENID-SAI, INIFAP) reported the draft genomes of two Mexican strains of hemoplasmas that infect cattle: *M. wenyonii* INIFAP02 and ‘*Ca*. M. haemobos’ INIFAP01. This whole-genome shotgun project was deposited at DDBJ/ENA/GenBank under accession no. LWUJ00000000 and QKVO01000000. Both genomes were de novo assembled using the SPAdes version 3.11.1 program.

The 12 hemoplasma genomes (two Mexican strains and ten strains from other countries) that infect different hosts, included in this study and reported in the GenBank database (https://bit.ly/314fOre, accessed on 20 August 2022, are listed in Appendix A. The general features of the 12 hemoplasma genomes were obtained using the quality assessment tool for genome assemblies (QUAST) (v5.0.2) program [21] with default settings. All genomes were annotated automatically to predict the coding sequences (CDS) using the rapid annotation using subsystem technology (RAST) (v2.0) server (https://bit.ly/2XjTTey, accessed on 22 August 2022) [22] with the classic RAST algorithm.

The mapping of ribosomal genes (rRNA) was performed based on the information reported in the NCBI database of genomes of *M. wenyonii* Massachusetts, *M. wenyonii* INIFAP02, and *Ca*. M. haemobos INIFAP01 (NC_018149.1; NZ_QKVO00000000.1; and LWUJ00000000.1, respectively). Transfer (tRNA) of genes was carried out using ARAGORN (v1.2.38) (https://bit.ly/3k1R2QT, accessed on 23 August 2022) server [23]. The sequence and length of 16S and 23S rRNA genes were obtained from the RNAmmer (v1.2) (https://bit.ly/3glQKCj, accessed on 25 August 2022) server [24].

### 2.2. Phylogenetic and Pangenome Analysis

For the phylogenetic reconstruction, we used 39 16S rRNA gene sequences that correspond to 12 16S rRNA gene sequences reported in the 12 hemoplasma genomes, including the two Mexican strains, and 3 16S rRNA gene sequences reported for bovine hemoplasmas whose genomes have not been reported. We also included 22 16S rRNA gene sequences from hemoplasmas that do not infect bovines, but other animals such as swine, alpaca, felines, sheep, goats, and others. As an outgroup, we used two 16S rRNA gene sequences from *Ureaplasma* spp. of the family *Mycoplasmataceae* that infects humans. In total, 39 16S rRNA gene sequences were used in the phylogenetic reconstruction.

All 16S rRNA gene sequences were obtained from the GenBank database (https://bit.ly/314fOre, accessed on 15 August 2022) using the nucleotide BLAST (Blastn) suite (https://bit.ly/3k2Wkvs, accessed on 16 August 2022) [25]. Multiple alignments between 39 16S rRNA gene sequences were made using the MUSCLE (v3.8.31) program [26]. The jModelTest (v2.1.10) program [27] was used to select the best model of nucleotide substitution with the Akaike information criterion. The phylogenetic tree was estimated under the maximum likelihood method using the PhyML (v3.1) program [28] with 1000 bootstrap replicates. The phylogenetic tree was visualized and edited using the FigTree (v.1.4.4) program (https://bit.ly/39ROMXV, accessed on 23 August 2022).

Two pangenome analyzes were performed using the GET_HOMOLOGUES (v3.3.2) software package [29] with the following options: (i) among the 12 hemoplasma genomes, and (ii) among the 3 genomes of bovine hemoplasmas. The Fasta amino acid (FAA) annotation files of hemoplasma genomes were used as input files by the GET_HOMOLOGUES software package. The get_homologues.pl and compare_clusters.pl Perl scripts were used to compute a consensus pan-genome, resulting from clustering the all-against-all protein BLAST (Blastp) results with the COGtriangles and OMCL algorithms. The pan-genomic analysis was performed using the binary (presence–absence) matrix.

### 2.3. Comparative Genomics

The average nucleotide identity (ANI) values of 12 hemoplasma genomes were calculated using the calculate_ani.py Python script (https://bit.ly/2X96hho, accessed on 27 August 2022) with the BLAST-based ANI (ANIb) algorithm. Ultimately, the level of conserved genomic sequences of bovine hemoplasmas was visualized by the alignment of the genomes of the Mexican strains (‘*Ca*. M. haemobos’ INIFAP01 and *M. wenyonii* INIFAP02) against the reference genome of *M. wenyonii,* Massachusetts, using the NUCmer program of MUMmer (v3.0) software package [30] to obtain the positions of nucleotides that were aligned, and Circos (v0.69-9) software package [31]. The circular comparative genomic map of bovine hemoplasmas was edited with Adobe Photoshop CC (v14.0 x64).

### 2.4. Prediction of Antigenic Proteins

After RAST annotation, we identified several proteins of the six subsystems: virulence, disease and defense, cell division and cell cycle, fatty acids, lipids and isoprenoids, regulation and cellular signaling, stress response, and DNA metabolism.

For *Ca*. M. haemobos INIFAP01, the proteins were selected from three subsystems including virulence, disease and defense, cell division and cell cycle, and stress response. For *M. wenyonii* INIFAP02, the subsystems selected were virulence, disease and defense, cell division and cell cycle, fatty acids, and lipids and isoprenoids. The selection of the subsystems was based on the fact that these were proteins with the potential to be antigenic, as shown in an initial antigenicity prediction with SVMTrip (http://sysbio.unl.edu/SVMTriP/, accessed on 20 August 2022). Finally, we selected 11 protein sequences of *M. wenyonii* INIFAP02 and 12 proteins of *Ca*. M. haemobos INIFAP01 that were submitted to the VaxiJen v2.0 server to predict protective antigens (http://www.ddgpharmfac.net/vaxijen/VaxiJen/VaxiJen.html, accessed on 20 August 2022) with default parameters.

### 2.5. Prediction of Subcellular Localization and Stability of Proteins

Predicted antigenic proteins of *M. wenyonii* INIFAP02 and *Ca*. M. haemobos INIFAP01 were submitted to predict the secondary structure server Raptor X (http://raptorx.uchicago.edu/, accessed on 21 August 2022).

### 2.6. Linear B-Cell Epitope Prediction and Three-Dimensional Modeling

B-cell epitopes of *M. wenyonii* INIFAP02 and *Ca*. M. haemobos INIFAP01 were predicted using BCEpred (https://webs.iiitd.edu.in/raghava/bcepred/bcepred_submission.html, accessed on 21 August 2022) predicts based on physicochemical properties such as hydrophilicity, flexibility, accessibility, turns, exposed surface, polarity, and antigenic propensity; and SVMTrip (http://sysbio.unl.edu/SVMTriP/, accessed on 22 August 2022), which predicts based on protein surface regions that are preferentially recognized by antibodies.

The PHYRE2 server was used to predict the tridimensional structure of the proteins of both hemoplasmas. Phyre2 PDB files were visualized with the Protter tool.

## 3. Results

### 3.1. General Features of Genomes

Of the twelve hemoplasma genomes, two genomes assembled as contigs (*Ca*. M. haemobos INIFAP01 and *M. wenyonii* INIFAP02) and ten genomes as a single chromosome. The features of the 12 hemoplasma genomes are shown in Table 1.

The genomic features of the hemoplasmas classify them into two groups. Group 1 (previously named *Haemobartonella*) comprises genomes of ‘*Ca*. M. haemobos’, *M. haemocanis*, and *M. haemofelis* species with a length from 0.9 to 1.1 Mb, coding sequences (CDS) from 1180 to 1650, and 31 tRNA genes. Group 2 (previously named *Eperythrozoon*) comprises genomes of ‘*Ca*. M. haemolamae’, ‘*Ca*. M. haemominutum’, *M. ovis, M. parvum, M. suis*, and *M. wenyonii* species with a length from 0.5 to 0.7 Mb, CDS from 578 to 1045, and 32 or 33 tRNA genes.

The mapping of rRNA genes shows that hemoplasmas are separated into two groups. The four genomes of group 1 contain one copy of the 16S–23S–5S rRNA operon (Figure 1A). The 16S rRNA gene sequence length of group 1 ranges from 1429 to 1486 bp. Conversely, seven genomes of group 2 contain one copy of the 16S rRNA gene ranging from 1469 to 1508 pb, separated from one copy of the 23S–5S rRNA operon (Figure 1B,C). In addition, the genome of *M. ovis* Michigan of group 2 contains two copies of the 16S rRNA gene with lengths of 1479 and 1493 bp, separated from each other and separated from one copy of the 23S–5S rRNA operon (Figure 1D).

The 16S rRNA gene sequence of ‘*Ca*. M. haemobos’ INIFAP01 has 82–98% alignment coverage and an identity of 98.71–99.93% with ‘*Ca*. M. haemobos’, ‘*Ca*. M. haemobos’ clone 307, ‘*Ca*. M. haemobos’ clone 311, and ‘*Ca*. M. haemobos’ isolate cattle no. 18. Additionally, the 16S rRNA gene sequence of ‘*Ca*. M. haemobos’ INIFAP01 has alignment coverage of 99% and an identity of 81.83 and 81.73% with *M. wenyonii* INIFAP02 and *M. wenyonii* Massachusetts, respectively.

On the other hand, the genomes of *M. wenyonii* INIFAP02 and *M. wenyonii* Massachusetts are very similar in length and G + C content to each other, and they have the same number of tRNA genes and distribution of rRNA genes. However, the 16S rRNA gene sequence of *M. wenyonii* INIFAP02 has an alignment coverage of 100% and an identity of 97.57% with *M. wenyonii* Massachusetts. Additionally, the 16S rRNA gene sequence of *M. wenyonii* INIFAP02 has: (i) alignment coverage of 91–98% and an identity of 99.24–99.93% with *M. wenyonii* isolate Fengdu, *M. wenyonii* clone 1, *M. wenyonii* isolate ada1, and *M. wenyonii* isolate C124; and (ii) alignment coverage of 90–98% and an identity of 97.50–97.87% with *M. wenyonii* strain CGXD, *M. wenyonii* isolate B003, *M. wenyonii* isolateC031, and *M. wenyonii* strain Langford.

### 3.2. Phylogenetic and Pangenome Analyzes

The model of nucleotide substitution of the phylogenetic tree based on the 16S rRNA gene of hemoplasmas was GTR + I + G. The phylogenetic tree shows that the hemoplasmas arrange into two groups (Figure 2). Group 1 (blue lines) contains two sub-groups: (i) ‘*Ca*. M. haemobos’ species; and (ii) *M. haemocanis* and *M. haemofelis* species. Group 2 (red lines) groups into two subgroups: (i) ‘*Ca*. M. haemolamae’, ‘*Ca*. M. haemominutum’, *M. ovis,* and *M. wenyonii* species; and (ii) *M. parvum* and *M. suis* species. The phylogenetic tree topology shows a divergence between the two subgroups of hemoplasmas.

Pangenome analysis among the 12 hemoplasmas shows that the core, softcore, shell, and cloud genomes are composed of 110, 146, 787, and 3099 gene clusters, respectively (Figure 3 and Appendix A). Additionally, the core genomes of groups 1 and 2 of hemoplasmas are composed of 236 and 149 gene clusters, respectively. Pangenome analysis among the three genomes of bovine hemoplasmas shows that the core genome comprises 154 gene clusters. Additionally, the two genomes of *M. wenyonii* species share 273 gene clusters. Moreover, ‘*Ca*. M. haemobos’ INIFAP01, *M. wenyonii* INIFAP02, and *M. wenyonii* Massachusetts contain 312, 190, and 157 unique gene clusters, respectively.

### 3.3. Comparative Genomics

ANIb values between different hemoplasma species show that the alignment coverage is less than 79% (Figure 4A and Appendix A), and the identity is less than 83% (Figure 4B and Appendix A). Moreover, ANIb values show that ‘*Ca*. M. haemobos’ INIFAP01 has an alignment coverage of 0.46 and 0.31%; and an identity of 74.12 and 74.16% with *M. wenyonii* INIFAP02 and *M. wenyonii* Massachusetts, respectively. Moreover, ANI values between the same species show that: (i) *M. haemofelis* Langford 1 and Ohio 2 genomes have an alignment coverage and identity of 97.65 and 97.41%, respectively; (ii) *M. suis* KI3806 and Illinois genomes have an alignment coverage and identity of 95.13 and 97.63%, respectively; and (iii) *M. wenyonii* Massachusetts and INIFAP02 genomes have an alignment coverage and identity of 51.58 and 79.37%, respectively. The ‘*Ca*. M. haemobos’ INIFAP01 genome only has three small regions (red lines highlighted with a green marker in the inner track) greater than 78% identity aligned with the *M. wenyonii* Massachusetts genome (the black circle in the outer track). Furthermore, the circular map shows that *M. wenyonii* INIFAP02 has few regions (blue lines in the intermediate track) greater than 78% identity aligned with the *M. wenyonii* Massachusetts genome (Appendix A).

### 3.4. Selection and Prediction of B-Cell Epitopes in Proteins

After the hemoplasmas genomes were annotated in the RAST server, only 18% of *Ca*. M. haemobos INIFAP01 proteins are classified in the different subsystems of RAST and 22% proteins for *M. wenyonii.* This fact exhibits a low percentage of known proteins in this database. For this reason, we manually reviewed the annotated proteins in the subsystems of both hemoplasmas. Then we selected those proteins whose category suggests a protein with potential antigenicity. The sequences of 11 proteins of *M. wenyonii* and 12 proteins of *Ca*. M. haemobos were submitted to VaxiJen, and the prediction of antigen/non-antigen for each selected protein was calculated (Table 2). The tridimensional structure of the proteins was performed to locate the predicted epitopes.

The collection of B-cell epitopes (linear antigens) was predicted with SVMTrip, and BCEPred server for those proteins predicted as “Antigen” by VaxiJen (Table 3).

## 4. Discussion

Hemoplasmas have constantly been detected in cattle in the last few years [32,33,34,35,36]. However, their transmission by ticks lacks evidence. On the contrary, their mechanical transmission mediated by blood-sucking flies and contaminated veterinary instruments could be essential for hemoplasmas dispersion [37,38]. In this scenario, information about hemoplasmas contributes to a better understanding of these pathogens and their impact on animal health. In this regard, we performed comparative genomic and immune-informatics studies of several hemoplasmas genomes reported worldwide.

This study aimed to provide information about hemoplasmas derived from a comparative genomic and how these data, along with immuno-informatics, provided potential antigenic candidates with the potential to be used in detection and diagnosis methods. This work comprises an analysis of the hemoplasmas reported in the last few years, including two Mexican strains, representing the most recent information in the field.

Hemoplasmas have undergone phylogenetic reclassification after several studies based on molecular markers [39]. Their genome size variation, positional shuffling of genes, and poorly conserved gene synteny is evidence of the high dynamics of their genomes [1]. All the genomic differences of the Mexican bovine hemoplasmas confirm this dynamic. In group 1, *Ca*. M. haemobos’ INIFAP01, we found the canonical structure of the rRNA operon, 16S–23S–5S; however, in group 2 *M. wenyonii* INIFAP02, we identified an unlinked rRNA gene structure (Figure 1). The canonical structure in bacteria allows a rapid response to the demand of changing growth conditions, whereas unlinked rRNA genes result in the genome degradation typical of obligate intracellular lifestyles, as reported in members of the order of Rickettsiales [40,41,42].

We also present 12 genomes of hemoplasmas classified into two groups (groups 1 and 2) and with different numbers of CDS and tRNAs. Additionally, the distribution of rRNA genes is specific to each species. Specifically, the genome of ‘*Ca*. M. haemobos’ INIFAP01 is significantly longer than the two genomes of *M. wenyonii* species, but ‘*Ca*. M. haemobos’ INIFAP01 has a lower G + C content. Again, these features denote the substantial gene gain/loss throughout the evolution observed in hemoplasmas [1].

The phylogenetic reconstruction shows that ‘*Ca*. M. haemobos’ INIFAP01 is phylogenetically distant through evolution from *M. wenyonii* INIFAP02 and *M. wenyonii* Massachusetts, whereas the strains INIFAP02 and Massachusetts are closely related through the evolution of group 2.

The number of genes in the core (those present in all considered genomes); softcore (found in 95% of genomes); shell (present in more than two genomes and less than 95% of genomes); and cloud genes (found in not more than two genomes) revealed in the pan-genomic analysis suggest that there is considerable loss/gain in genes through the evolution of the 12 hemoplasmas genomes.

Regarding the pangenome analysis of bovine hemoplasmas, the low number of gene clusters in the core genome confirms that ‘*Ca*. M. haemobos’ INIFAP01 is a divergent species from *M. wenyonii* INIFAP02 and *M. wenyonii* Massachusetts. In comparative genomics of bovine hemoplasmas, low percentages of alignment coverage and identity of *Ca*. M. haemobos INIFAP01, *M. wenyonii* INIFAP02, and *M. wenyonii* Massachusetts in the ANI values suggest that the *Ca*. M. haemobos INIFAP01 genome has a different structure when compared with the genomes of *M. wenyonii* INIFAP02 and *M. wenyonii* Massachusetts.

Surprisingly, the alignment coverage and identity percentages among both genomes of *M. wenyonii* (strains INIFAP02 and Massachusetts) suggest that these strains may not belong to the same species because the ANI values are <95%, the species ANI cutoff value [43,44,45].

Since bovine hemoplasmas show significant differences at the genomic level, and their impact on cattle health causes economic losses, we performed an immune-informatic analysis to identify B-cell epitopes that could be used to design molecular detection methods of bovine anaplasmosis. The immune-informatics studies predict several B-cell epitopes (Table 3), which could be applied in molecular detection methods and vaccines. Peptides that contain epitopes are widely applied for pathogen detection by serological methods [46,47], immunolocalization of pathogen proteins [48], and vaccines against animal diseases [49,50,51]. For *Ca*. M. haemobos INIFAP01, the proteins predicted with antigenic peptides are DNA gyrase subunit B, SSU ribosomal protein S7p and S12p, and translation elongation factor G. All of them participate in vital processes. We observe something similar in *M. wenyonii* INIFAP02, where the proteins predicted with antigens also participate in vital processes (SSU ribosomal protein S7p and S12p, translation elongation factor G, ribosomal protein LSU L35p) (Table 3).

The epitope collection generated in this work will help to design molecular tools to diagnose bovine hemoplasmosis. Undoubtedly, this approach could have further applications as part of the prevalence of studies with animals from different geographic regions. Also, this strategy should consider the best alternative of the antigenic peptide to discriminate from other pathogens that infect bovines.

## 5. Conclusions

This work describes the evolutionary relationships between 39 hemoplasmas reported until now, including genomes, isolates, and clones. Also, we present the main genomic characteristics and pangenome analyses of twelve hemoplasma genomes and a prediction of B-cell epitopes in proteins annotated in *M. wenyonii* INIFAP02 and *Ca*. M. haemobos INIFAP01. Regarding *Ca*. M. haemobos INIFAP01, the proteins predicted with antigenic potential that include components of vital processes (DNA gyrase subunit B, SSU ribosomal protein S7p and S12p, translation elongation factor G) are the candidates to initiate studies for molecular detection of this hemoplasma. We observe something similar in *M. wenyonii* INIFAP02, where the predicted antigens also participate in vital processes (SSU ribosomal protein S7p and S12p, translation elongation factor G, ribosomal protein LSU L35p). Finally, the data presented here about antigenic peptides of *M. wenyonii* INIFAP02 and *Ca*. M. haemobos INIFAP01, identified by immuno-informatics, have potential use in detecting and preventing hemoplasmosis.

## Figures and Tables

**Figure 1 microorganisms-10-01916-f001:**
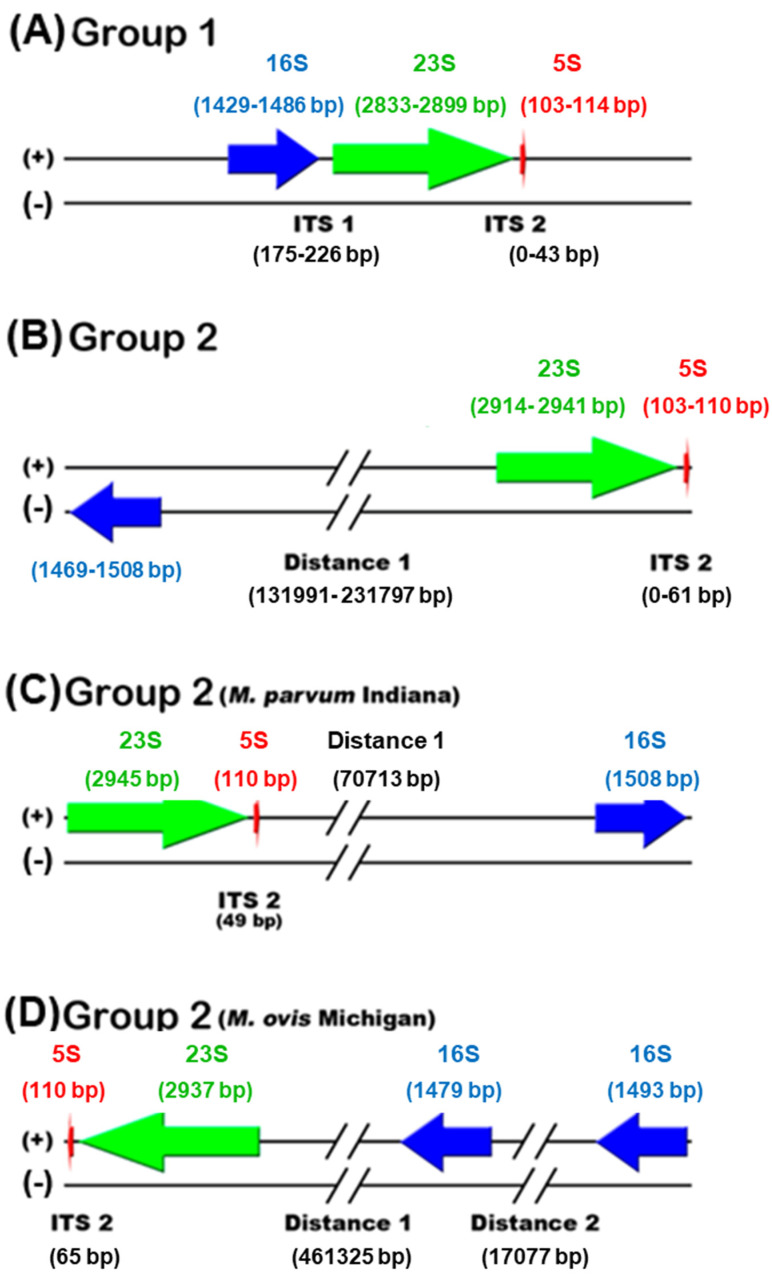
Mapping of rRNA genes of hemoplasmas. (**A**) Genomes of ‘*Ca*. M. haemobos’ INIFAP01, *M. haemocanis* Illinois, *M. haemofelis* Langford 1, and *M. haemofelis* Ohio 2 of group 1 contain one copy of 16S–23S–5S rRNA operon. (**B**) Genomes of ‘*Ca*. M. haemolamae’ Purdue, ‘*Ca*. M. haemominutum’ Birmingham 1, *M. suis* Illinois, *M. suis* KI3806, *M. wenyonii* INIFAP02, and *M. wenyonii* Massachusetts of group 2 contain one copy of 16S rRNA gene, which is separate from one copy of 23S–5S rRNA operon in a different chain. (**C**) Genome of *M. parvum* Indiana of group 2 contains one copy of 16S rRNA gene, which is separate from one copy of 23S–5S rRNA operon in the same chain. (**D**) Genome of *M. ovis* Michigan of group 2 contains two copies of 16S rRNA gene, which are separated from each other, and they are separated from the one copy of 23S–5S rRNA operon in the same chain. The 16S, 23S, and 5S rRNA genes are represented by blue, green, and red arrows, respectively.

**Figure 2 microorganisms-10-01916-f002:**
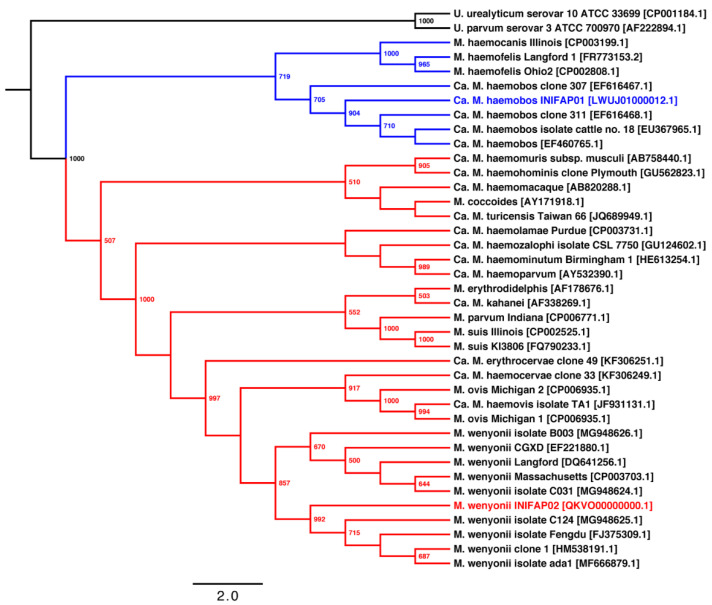
Phylogenetic relationships based on the 16s rRNA genes of group 1 (blue lines) and 2 (red lines) of hemoplasma species. The phylogenetic tree was obtained using the PhyML program with the maximum likelihood method and 1000 bootstrap replicates. Bootstrap values (>50%) are displayed in the nodes. The model of nucleotide substitution was GTR + I + G. The INIFAP01 and INIFAP02 Mexican strains of bovine hemoplasmas are shown in blue and red letters, respectively. GenBank accession numbers are shown in square brackets.

**Figure 3 microorganisms-10-01916-f003:**
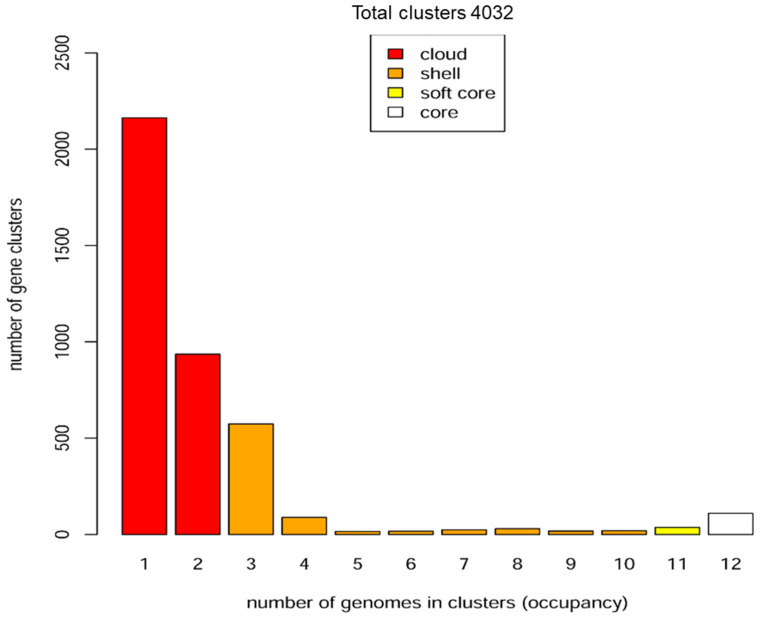
Pan-genome distribution in four categories (cloud, shell, soft core, and core) of 12 hemoplasmas. The core, softcore, shell, and cloud genomes are composed 110, 146, 787, and 3099 gene clusters, respectively.

**Figure 4 microorganisms-10-01916-f004:**
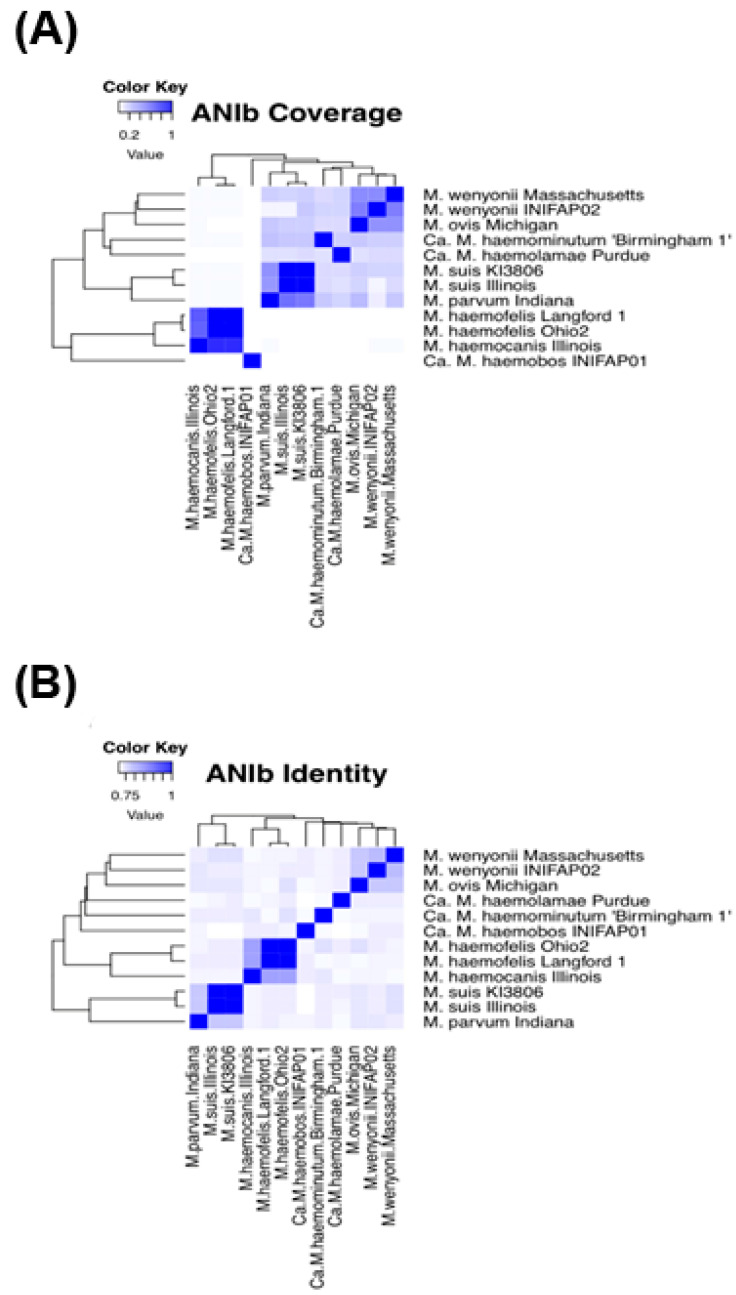
Heatmaps of blast-based average nucleotide identity (ANIb) values of 12 hemoplasma genomes. (**A**) Heatmap of ANIb values of alignment coverage. (**B**) Heatmap of ANIb values of identity. Color intensity increases from white to deep blue when ANIb values approach from 0.0 to 1.0 (0–100%), respectively. The more intense the color, the greater the coverage and identity percentages.

**Table 1 microorganisms-10-01916-t001:** General features of 12 hemoplasma genomes.

Group	Organism	Assembly Level	Length (bp) *	G + C Content (%) *	CDS **	rRNAs ^#^	tRNAs ^##^
Group 1	‘*Ca.* M. haemobos’ INIFAP01	18 contigs	935,638	30.46	1180	3	31
Group 1	*M. haemocanis* Illinois	Chromosome	919,992	35.33	1234	3	31
Group 1	*M. haemofelis* Langford 1	Chromosome	1,147,259	38.85	1595	3	31
Group 1	*M. haemofelis* Ohio2	Chromosome	1,155,937	38.81	1650	3	31
Group 2	‘*Ca.* M. haemolamae’ Purdue	Chromosome	756,845	39.27	1045	3	33
Group 2	‘*Ca.* M. haemominutum’ Birmingham 1	Chromosome	513,880	35.52	587	3	32
Group 2	*M. ovis* Michigan	Chromosome	702,511	31.69	918	4	32
Group 2	*M. parvum* Indiana	Chromosome	564,395	26.98	578	3	32
Group 2	*M. suis* Illinois	Chromosome	742,431	31.08	914	3	32
Group 2	*M. suis* KI3806	Chromosome	709,270	31.08	856	3	32
Group 2	*M. wenyonii* INIFAP02	37 contigs	596,665	33.43	678	3	32
Group 2	*M. wenyonii* Massachusetts	Chromosome	650,228	33.92	727	3	32

CDS: coding sequences; * data obtained with the QUAST program; ** data obtained with the RAST server; ^#^ data obtained with the RNAmmer server; and ^##^ data obtained with the ARAGORN server.

**Table 2 microorganisms-10-01916-t002:** Prediction of antigenicity of proteins of *Ca*. M. haemobos and *M. wenyonii*.

*Candidatus* Mycoplasma Haemobos INIFAP01	
Classification (NCBI Accession Number)	Prediction Score as Antigen (VaxiJen)
RAST Category: Virulence, disease, and defense	
DNA gyrase subunit B (OAL10308.1)	0.5352 (antigen)
DNA gyrase subunit A (OAL10309.1)	0.4373 (non-antigen)
SSU ribosomal protein S7p (WP_187150158.1)	0.5180 (antigen)
Translation elongation factor G (WP_187150159.1)	0.5399 (antigen)
Translation elongation factor thermo unstable (Tu) (WP_187150070.1)	0.4268 (non-antigen)
SSU ribosomal protein S12p (WP_187150157.1)	0.7537 (antigen)
DNA-directed RNA polymerase beta subunit (WP_187150197.1)	0.3781 (non-antigen)
DNA-directed RNA polymerase (WP_187150196.1)	0.4488 (non-antigen)
RAST Category: Division and cell cycle	
ProteinTsaD/Kae1/Qri7 (WP_187150270.1)	0.3846 (non-antigen)
RNA polymerase sigma factor RpoD (WP_187150278.19	0.3857 (non-antigen)
DNA primase (WP_187150493.1)	0.3009 (non-antigen)
RAST Category: Fatty acids, lipids and isoprenoids	
Cardiolipin synthase (WP_187150134.1)	0.3463 (non-antigen)
RAST Category: Stress Response	
Manganese superoxide dismutase (WP_187150149.1)	0.3487 (non-antigen)
***Mycoplasma wenyonii* INIFAP02**	
RAST Category: Virulence, disease, and defense	
Ribosomal protein SSU S7p (RAO94848.1)	0.5435 (antigen)
Translation elongation factor G (RAO94847.1)	0.5650 (antigen)
Translation elongation factor thermo unstable (Tu) (RAO95121.1)	0.4359 (non-antigen)
Ribosomal protein SSU S12p (RAO94849.1)	0.7774 (antigen)
Ribosomal protein LSU L35p (RAO95358.1)	0.5491 (antigen)
Translation initiation factor 3 (RAO95223.1)	0.4488 (non-antigen)
Ribosomal protein LSU L20p (RAO95357.1)	0.3668 (non-antigen)
RAST Category: Division and cell cycle	
Protein TsaD/Kae1/Qri7 (RAO95106.1)	0.3954 (non-antigen)
RNA polymerase sigma factor RpoD (RAO94807.1)	0.3979 (non-antigen)
DNA primase (RAO95339.1)	0.3929 (non-antigen)
RAST Category: Fatty acids, lipids and isoprenoids	
Cardiolipin synthase (RAO95377.1)	0.3305 (non-antigen)

**Table 3 microorganisms-10-01916-t003:** B-cell epitopes of *Ca*. M. haemobos and *M. wenyonii* predicted by immunoinformatics.

*Ca.* M. Haemobos INIFAP01			*M. wenyonii* INIFAP02		
RAST Category: Virulence, disease, and defense	Predicted epitopes (SVMTrip, recommend score 1.0)	Predicted epitopes (BCEPred)	RAST Category: Virulence, disease, and defense	Predicted epitopes (SVMTrip, recommend score 1.0)	Predicted epitopes (BCEPred)
DNA gyrase subunit B	**RKLALEGFMSFAGKLADCTT**	AGGDSSDSGGQYTDS **GGKFDNNSYKTSGG *** EVNVYRNGEEHY **ENGGKIKDEPKMVSKCEEDKTG** IESRLTKLAYLNKGKKFV **VNEITKEEKEFFYEEGIKDW** FIHSEGKVKNRRAPE **FGRFLEENPEQRKVILQRVDQERNFRLK** VVEGDSAGGSAKSARNREYQAI **NVWKRSKYTAILENEEVKSL** NKEVVYLFDDKKKDEFLKNLSNP	Ribosomal protein SSU S7p	** MWEGKKQLARRIVYNALEKI **	NALEKIREKTEKNPVEV **YQVPVESSKERREALA LIKYSRKRN**
SSU ribosomal protein S7p	**PLEVFMEALKNIAPTIELKT**		Translation elongation factor G	IPKEYIKSIREGLVDAMKAG **VPRIIFCNKMDKVGASFQSS**	DAGKTTTSERDWMEQEREKGITDEEFEEIPI **PEDQQEEVKTLR** KAFTRSGEELTIENKDESN
Translation elongation factor G	**EFVDKIVGGKIPKEYIKSIK** AKVIKSKIPLKEMFGYATAL	DWMEQEKEKGIT **TKKAYEFDGKQEEEYKEIPI** ETPAFDKEQNPISIKNSPDNDF **QMHSNHRTEIES** AIEPKTKVDQEKMSM **FRETFTQEAEVEGKYIKQSGGRG** HVWIKYEPNKDKGFEF **LSLKDASKKCASILLEPI** SRRGTIEGDEQVENAKVIKSKIPLKEM	Ribosomal protein SSU S12p	** RVKDLPGVKYHIIRGKLDAA **	VEKRKKERSKYGVKKEKKS
SSU ribosomal protein S12p	**RVKDLPGVKYHIVRGKLDTV**		Ribosomal protein LSU L35p	** SHRSHCASAKTTKRKRQLRK **	**KKIKHKTKKSLSKR** SGAIKRKRSHRS **SGAIKRKRSHRSHCASAKTTKRKRQLRKSA**

* To distinguish the complete sequences of the epitopes they are shown as bolded.

## Data Availability

The genomic data used to support the findings of this study were deposited in the GenBank repository with accession numbers NZ_QKVO00000000.1 and LWUJ00000000.1.

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
