# Peer review of "An Update of Bovine Hemoplasmas Based on Phylogenetic and Genomics Analysis"

_microorganisms, 2022, doi:10.3390/microorganisms10101916_

Round 1
Reviewer 1 Report
The work focused on genomic characterization of a wide series of hemoplasmas, several of which directly involved in animal infection process. Phylogenetic analysis can improve the knowledge about this group of pathogens, opening further possibilities for preventive strategies. Several additional information and data obtained from the performed analyses can help in clarifying some characteristics of those particular microrganisms
Author Response
"Please see the attachment."

Reviewer 2 Report
The paper “An update of bovine hemoplasmas based on phylogenetic and genomics analysis” is very interesting, specially because these bacteria are unculturable. The work is well designed and executed. The paper is well written and the results are very interesting. The novelty of this paper is in the information about hemoplasmas derived from a comparative genomic and how this data, along with immuno-informatics, provided potential antigenic candidates with the possible potential to be used in detection and diagnosis methods.
Specific comments:
Through the paper, put latin words (in vitro L. 31, de novo L. 77) and class, order, family names in cursive (Mollicutes L. 30, Anaplasmataceae L. 32, Rickettsiales L. 32 and 351, Mycoplasma L. 34).
L. 38-39: Mycoplasma (M.) wenyonii strains: Massachusetts and INIFAP02, and one Candidatus (Ca.) Mycoplasma (M.) haemobos…, instead of Mycoplasma wenyonii strains: Massachusetts and INIFAP02, and one Candidatus Mycoplasma haemobos…
L. 163: I do not see the tridimensional structure of the proteins in the results.
L. 320: B-Cell epitopes instead of epitores
L. 336-341: Discussion about protein predictions should go after the genomic analysis as in the material and methods and the results.
Suplementary data:
Titles of tables S2 and S3 are exactly the same. Add a longer explanation to give more information.
Author Response
"Please see the attachment"
